# Bilateral Sensorimotor Cortical Communication Modulated by Multiple Hand Training in Stroke Participants: A Single Training Session Pilot Study

**DOI:** 10.3390/bioengineering9120727

**Published:** 2022-11-24

**Authors:** Jian-Jia Huang, Yu-Cheng Pei, Yi-Yu Chen, Shen-Shiou Tseng, Jen-Wen Hung

**Affiliations:** 1Department of Physical Medicine and Rehabilitation, Chang Gung Memorial Hospital, Linkou, Taoyuan City 333, Taiwan; 2Master of Science Degree Program in Innovation for Smart Medicine, Chang Gung University, Taoyuan City 333, Taiwan; 3Center of Vascularized Tissue Allograft, Chang Gung Memorial Hospital, Linkou, Taoyuan City 333, Taiwan; 4Department of Rehabilitation, Chang Gung Memorial Hospital-Kaohsiung Medical Center, Kaohsiung 833, Taiwan; 5Department of Physical Medicine and Rehabilitation, Tao-Yuan General Hospital, Taoyuan City 330, Taiwan; 6School of Medicine, College of Medicine, Chang Gung University, Taoyuan City 333, Taiwan

**Keywords:** robot-assisted training, stroke, mirror therapy, bimanual training, EEG, coherence, sensorimotor cortex

## Abstract

Bi-manual therapy (BT), mirror therapy (MT), and robot-assisted rehabilitation have been conducted in hand training in a wide range of stages in stroke patients; however, the mechanisms of action during training remain unclear. In the present study, participants performed hand tasks under different intervention conditions to study bilateral sensorimotor cortical communication, and EEG was recorded. A multifactorial design of the experiment was used with the factors of manipulating objects (O), robot-assisted bimanual training (RT), and MT. The sum of spectral coherence was applied to analyze the C3 and C4 signals to measure the level of bilateral corticocortical communication. We included stroke patients with onset <6 months (n = 6), between 6 months and 1 year (n = 14), and onset >1 year (n = 20), and their Brunnstrom recovery stage ranged from 2 to 4. The results showed that stroke duration might influence the effects of hand rehabilitation in bilateral cortical corticocortical communication with significant main effects under different conditions in the alpha and beta bands. Therefore, stroke duration may influence the effects of hand rehabilitation on interhemispheric coherence.

## 1. Introduction

Stroke is a major neurological disorder that causes sensorimotor dysfunctions. After stroke, functional recovery in the upper limb is slower than that in the lower limb, thus posing a challenge [1], particularly for patients with chronic stroke. In addition, hand function is complex, and multiple abilities must be integrated to perform a specific task in daily life, such as grasping, pinching, or gripping [2,3,4]. Task-oriented training [5,6], bimanual training (BT) [7], mirror therapy (MT) [8], and robot-assisted therapy [9] have demonstrated benefits for the functional recovery of hand function in stroke patients. In addition, combination therapies such as robots with BT or MT have also been suggested in clinical practice [10,11]. However, the mechanism of action during training remains unclear.

Brain activity is modulated by rehabilitation training and may contribute to functional recovery. For example, BT modulates the motor cortex by disinhibiting cortical activity and involving other pathways to facilitate cortical plasticity [12]. At the same time, MT decreases cortical inhibition in the ipsilateral primary motor cortex or increases the cross-education effect [13,14]. A near-infrared spectroscopy study showed that robot-assisted BT for hand function training induced more balanced cross-hemispheric activities in stroke patients than MT or robotic passive movement therapy [9]. Thus, rehabilitation training may decrease inhibition in damaged brain areas.

EEG studies using event-related desynchronization (ERD) have shown that desynchronization between motor events and responding brain signals increased during active hand training [15,16,17,18]. Regarding functional imaging that investigated interhemispheric activation, Christian Grefke and Gereon R. Fink (2014) suggested that well-balanced bilateral cortical excitatory and inhibitory connections are required for optimizing sensorimotor function [19]. A transcranial magnetic stimulation (TMS)-EEG study of bilateral motor cortical coherence in the beta oscillatory band is considered to reflect GABAergic activity associated with interhemispheric inhibition required for skilled movement, and reduced beta interhemispheric inhibition may contribute to motor recovery [20]. Another study on bilateral sensory cortical interaction showed that interhemispheric coherence is reduced in relation to functional sensorimotor improvement after rehabilitation [21]. Although the studies provided conflicting results, they may imply that the increased bilateral sensorimotor neural activity may be related to functional recovery. However, the neural functions in each hemisphere may become independent. Thus, more evidence is needed to understand neural communication as participants receive training.

In the present pilot study, we investigated the communication between bilateral sensorimotor cortices in stroke participants who had a wide range of stroke onset times, while receiving unilateral hand movement, MT, or robot-assisted bimanual training (RT), and combined interventions with or without object manipulation (O). Non-invasive EEG recordings were conducted to record brain activity while participants were performing training tasks. Finally, coherence analysis was applied to study the interhemispheric connections. Specifically, a factorial design was applied, including conditions involving a mirror, robot, and object manipulation. We hypothesized that varying hand rehabilitation training conditions would alter bilateral communication.

## 2. Materials and Methods

### 2.1. Participants and Design

The inclusion criteria for the data of stroke patients used in the analysis were as follows: (1) age ≥ 20 years; (2) first-ever stroke patients diagnosed by clinical presentation and imaging findings; (3) no serious cognitive impairment and ability to follow orders and understand the study procedures; (4) modified Ashworth scale of finger flexors not more than 3; and (5) no visual field deficits or hemineglect. The exclusion criteria were as follows: (1) bilateral hemispheric or cerebellar lesions; (2) seizure disorder; (3) any fixed joint contracture of the affected upper limb; and (4) a history of other neurologic or neuromuscular diseases. We applied Brunnstrom recovery stages to evaluate functional motor recovery. The results of Brunnstrom recovery stages are divided into six stages, ranging from stage 1 (flaccidity) to stage 6 (spasticity disappears and individual joint movement is available), and have been widely applied in stroke patient assessment [22]. The experimental protocol was approved (IRB number: 201900537A3), and written informed consent was obtained from all participants. All methods were performed in accordance with the regulations of the Human Subjects Research Act and guidelines of the Declaration of Helsinki 1975.

### 2.2. Exoskeleton Hand Device

A wearable exoskeletal hand, Mirror Hand (Mirror Hand, HS001, Rehabotics Medical Technology Corporation, Taiwan), was used for the RT in this study. The Mirror Hand applies a master (unaffected hand)–slave (affected hand) mechanism for bilateral hand movements [9,10,23], a mechanism through which the fingers in the exoskeletal hand can be moved, mirroring the movement of the fingers in the unaffected hand. This is unlike previous upper-extremity robots that unilaterally train the affected hand [11,24]. The device consists of an exoskeletal hand, which has five individual exoskeleton finger modules, each of which provides external power to move the affected individual fingers to perform individual finger flexion and extension movements at a constant speed (approximately 3 s from extension to flexion, and vice versa), and a sensor glove, used to detect flexion/extension movements of the unaffected individual fingers and to control the exoskeletal hand via a control box, which was used to process the signal detected from the sensor glove and deliver the commands to move the exoskeletal hand (Figure 1A).

### 2.3. EEG Recording

EEG activity was recorded using a wearable dry electrode EEG recorder (DSI-24 system, San Diego, CA, USA) with 18 scalp channels and 2 references attached to the subject’s earlobes (Figure 2A). The signal was filtered from 1 Hz to 120 Hz, and the sampling rate of the recording was 300 Hz. Before the experiment, the quality of the EEG signal was tested under both eyes-open and eyes-closed conditions. The quality of the EEG signal is satisfied if alpha oscillatory (7–12 Hz) activity in the eyes-closed condition can be observed in the EEG signal and the oscillatory brain wave is suppressed in the eye-open condition. To investigate bilateral sensorimotor cortical activities, the recording channels overlying the primary sensorimotor and mesial motor areas, marked by C3 and C4, respectively, were used for analysis [25] (Figure 2B).

### 2.4. Experimental Design and Procedure

The present study used a 2 (with vs. without object manipulation) × 2 (with vs. without robot-assistance) × 2 (with vs. without mirror) factorial design experiment (Figure 1B). For each hand movement condition, the participants repeated the movements slowly for at least 60 s. One minute of rest was allowed between conditions when the EEG signals were not recorded. All training sessions were conducted in an individual room inside a general occupational therapy room in the hospital. EEG recordings were performed by a research assistant. The experiment was stopped when any discomfort or adverse events occurred, and adequate medical assistance was provided if necessary. Each training condition was applied once, and the sequence was randomized. The training conditions were as follows:

#### 2.4.1. Unilateral Free Hand Movement (Hand)

The participants were asked to flex/extend their affected fingers and place the other hand resting on a table (Figure 2C). Participants were instructed to complete the task while looking at their moving hands.

#### 2.4.2. Mirror Therapy (MT)

MT was performed using unilateral hand movements. The participants sat in front of a table, and a mirror was placed in the midsagittal plane beside one hand (or unaffected hand) to block the participant’s view of the other hand (or affected hand). The participants were instructed to look at the reflection of the unaffected hand in the mirror as if it were the affected hand. The participants were instructed to flex/extend their unaffected fingers and imagine the affected fingers moving as mirrored movements of the unaffected fingers (Figure 1D).

#### 2.4.3. Robot-Assisted Bimanual Therapy (RT)

The participants wore a sensor glove on the unaffected hand and a wearable exoskeleton hand on the affected hand (Figure 1E). The participants were instructed to flex/extend their unaffected fingers, and their affected fingers were passively moved via the exoskeletal hand.

#### 2.4.4. Robotic Therapy with Mirror Therapy (RT × MT)

Under these conditions, participants wore robotic hands to perform mirror therapy. The participants wore the exoskeleton hand system, and a mirror was placed between the two hands in the participant’s midsagittal plane. The participants flexed or extended their unaffected fingers with the sensor glove and observed the reflection in the mirror. Simultaneously, the fingers on the affected side were passively moved by the exoskeleton (Figure 1F).

#### 2.4.5. Object Manipulation (O)

The conditions marked with O indicate the addition of object manipulation, including the O (hand movement with object), RT × O, MT × O, and RT × MT × O conditions. In the RT × O and RT × MT × O conditions, the participants grasped/released a cup (the object) with their unaffected hand, while the affected hand performed the same task aided by the exoskeletal robot. Under the MT × O and MT × O conditions, the participant’s affected hand held the object (cup) without grasping/releasing movements (Figure 1G–J).

### 2.5. Data Analysis

All calculations were performed in MATLAB (MathWorks Inc., Natick, MA, USA) and Microsoft Excel.

#### 2.5.1. Coherence Sum

First, *coherence analysis* (Equation (1)) was used to analyze neural interactions between brain areas [21,26]. The EEG signal was recorded in each condition for 60 s and analyzed. The signals recorded from the C3 and C4 channels were selected as the regions of interest in the present study, which targeted the left- and right-side sensorimotor cortical areas. We analyzed three frequency bands: alpha band (7–12 Hz), low beta band (13–19 Hz), and high beta band (20–30 Hz) (Figure 3A,B). Coherence calculations were performed using the *mscohere* function in MATLAB (*mscohere* (*input data*, *reference*, *window length*, *number of overlaps*, *nFFT*, *Fs*) with nFFT = 256, overlap = 128, and Hanning window length = 256) with a frequency resolution of 1.1719 Hz (Figure 3C).
(1)$$C_{xy}\left(f\right)=\frac{{\left|P_{xy}\left(f\right)\right|}^{2}}{P_{xx}\left(f\right)P_{yy}\left(f\right)}$$
where $P_{xx}\left(f\right)$ and $P_{yy}\left(f\right)$ are the PSDs of channels C3 and C4, respectively, and $P_{xy}\left(f\right)$ is the cross PSD between these two channels; the coherence ($C_{xy}$) ranges from 0 to 1, and a coherence increment indicates greater interregional communication and vice versa. Next, each frequency band was acquired from the coherence spectrum and calculated using Equation (2), where f is the data number of the collected frequency band, and fb is the frequency band (Figure 3C–a).
(2)$$oherence\;sum=\overset{f}{\underset{i=1}{\sum }}C_{xy,\;fb}\left(f\right)\;$$

#### 2.5.2. Statistics

Statistical analyses were performed using SPSS software (IBM SPSS Statistics; IBM Corp., Armonk, NY, USA). The effect size was estimated by $\eta^{2}$; for example, $\eta^{2}$ = 0.1 implies that group membership can account for 10% of the total variance. Data are presented as mean ± standard deviation. In contrast, in the analysis of coherence sum, multiple-way repeated-measures analyses of variance (ANOVA) were performed using the onset ≤1 year vs. the onset >1 year stroke group as the between-group factor, and O, RT, and MT as the within-group factors. For the stroke group, we performed subgroup analysis, in which the factors for the hand side (right hemisphere vs. left hemisphere) and lesion site (cortical vs. subcortical) were analyzed. The *p*-value considered for statistical significance was set at *p* < 0.05.

## 3. Results

Data from 40 stroke participants were analyzed in the present study. The stroke onset time was 20.51 ± 23.41 months, ranging from 0.5 to 83 months. The distal UE Brunnstrom recovery stage ranged from two to four. Table 1 presents demographic data. No adverse events were reported during the study period.

The stroke onset time may have affected the results; therefore, we examined the effects of the EEG responses during training at different stroke onset times. We first examined whether interhemispheric coherence differed between the stroke onset ≤1 year (n = 20) and onset >1 year (n = 20) groups. The results showed significant between-group differences in the high-beta (*p* = 0.047, ⴄ^2^ = 0.1) bands but not in the alpha or low-beta bands. Figure 4 shows the results of the coherence sum of C3 and C4 in the alpha (Figure 4 top), low beta (Figure 4 middle), and high beta (Figure 4 bottom) bands in the two groups. We next tested the interaction effect (onset <6 months group, n = 6 vs. onset between 6 months and 1 year group, n = 14), and the results showed no significant difference in any condition.

As the two groups had significant differences in stroke onset times, they were separately analyzed in the following analyses. In each analysis of the onset ≤1 year group and onset >1 year group, we first examined the interaction effect of the damaged brain area (cortical vs. subcortical) and affected hand side (right hemisphere vs. left-hemispheric), respectively. If the interaction effect was significant, an additional subgroup was used for further analysis.

In the onset ≤1 year group, the three-way repeated-measures ANOVA showed no interaction effect in the alpha, low beta, or high beta bands. Although no significant main effect was found in the effects of MT, RT, and O factors, a significant main effect was observed on the RT × O condition in the high beta band (*p* = 0.01, ⴄ^2^ = 0.3) (Table 2 and Table 3). In the onset >1 year group, the three-way repeated-measures ANOVA showed no interaction effect in the alpha, low beta, or high beta bands. Next, we analyzed the main effects of the varied training conditions and found that the training with RT had significantly lower coherence than the without RT conditions in the high beta band (*p* = 0.04, ⴄ^2^ = 0.204); besides, in the high beta band, we also found the main effect in the condition of MT × RT × O (*p* = 0.048, ⴄ^2^ = 0.191), but no significant main effect in the other O conditions. We also found that the MT conditions had a significantly lower coherence sum in the alpha band (*p* = 0.012, ⴄ^2^ = 0.286), but not in the low or high beta bands. No significant main effect was found in the low-beta band (Table 4 and Table 5).

## 4. Discussion

In the present study, we enrolled patients with a wide range of post-stroke durations and found that the inter-hemispheric coherence sum is different between groups of post-stroke ≤1 year and >1 year across the alpha and high beta bands. MT and RT had significant effects on bilateral brain communication during the training.

Hand rehabilitation training strategies can be divided into those with and without object manipulation. Different types of training have been suggested to be served by distinct neural networks [27,28]. Object manipulation alone or combined with MT elicits a better therapeutic effect [29]. We found significant main effects in the high beta band at the RT × O and MT × RT × O conditions for patients with onset ≤1 year and >1 year, respectively; however, there was no main effect in the O condition. This finding could be attributed to the dysfunction of sensorimotor integration in stroke patients due to lesions in sensorimotor brain areas. Some patients with stroke may have proprioceptive dysfunction. In addition, post-stroke motor weakness prevents a normal proprioceptive experience because the affected UE has limited active motion. Thus, providing abundant proprioceptive experiences for stroke patients during rehabilitation training is recommended. RT × O and MT × RT × O provide more proprioceptive experiences than O, RT, and MT conditions, which may explain the significant main effects in the high beta band under the conditions of RT × O and MT × RT × O for patients with onset ≤1 year and >1 year, respectively.

Corticocortical neural oscillations of the sensorimotor areas in the alpha and beta bands are associated with voluntary movement and have different functions in the neural circuit [30,31]. These oscillations are related to the most prominent task-related spectral changes in the visual inspection of raw spectra [32] and finger movements [33,34,35]. Alpha-band neural oscillatory activity is thought to originate from the postcentral gyrus, and may contribute more to somatosensory activation. Beta-band neural oscillatory activity, which is considered to originate from the precentral gyrus, may participate in motor activation. An increase in coherence may reflect greater cooperation between neuronal assemblies; conversely, a decrease in coherence may reflect noncooperation. However, event-related desynchronization of the alpha and beta bands during movement has never been reported [15,16,17,18]. A desynchronized EEG may reflect that small neuronal assemblies work relatively independently or are desynchronized in the underlying neuronal circuitry. The event-related desynchronization phenomenon may support our results, as neural oscillatory coherence in the alpha and beta bands decreased under the MT and RT conditions. These findings imply that, first, the brain function participating in evoking motor activities may involve a combination of multiple bands; second, activating the affected hand motion through visual feedback, sensorimotor integration, and motor intention using MT and RT may facilitate neural function in the affected brain areas and decrease the interactions between cortical regions. However, direct evidence for this hypothesis should be obtained in the future.

Furthermore, Roushdy et al. indicated that motor recovery in stroke patients in the acute stage might be due to compensatory activity in the contralesional hemisphere [36]. We also found that the coherence sum of the RT × O condition in patients with post-stroke duration ≤1 year significantly increased, which may imply that bilateral sensorimotor cortical communication was enhanced under this operation. In the chronic stroke stage, the ipsilesional hemisphere may recover its function and become more active [36]. Conversely, Philips et al. found that beta band activities decreased in functional motor recovery [37]. We also found that, in patients with a post-stroke duration >1 year, the coherence sums at MT, RT, and MT × RT × O conditions decreased.

The question remains whether a decrease or increase in interhemispheric communication reflects a decrease in brain activity during movement. In a computational model, Manganotti et al. (1998) suggested that an increase or decrease in neural oscillatory coherence induced by task-related activity cannot differentiate between inhibitory and excitatory connectivity in the neural circuit, respectively [32]. Studies on interhemispheric connections using the TMS technique have also shown conflicting results regarding increasing and decreasing bilateral cortical interaction [38]. Further studies on interhemispheric communication at the molecular and neural circuitry levels in in-vivo animal models or humans are needed. A functional near-infrared spectroscopy (fNIRs) study recently showed that a single training session that included integrating visual, somatosensory feedback, and motor intention in hand training with robot-assisted BT could induce the greatest brain activity compared with unilateral MT or robot-assisted PROM [9], which supports our findings. MT with unaffected hand movement only induced interhemispheric activities in the sensorimotor areas. The training effects could also be observed by removing the intervention of the mirror, implying that robot-assisted BT could modulate sensorimotor activities in the cortex.

Although a single training session was not considered to induce neuroplasticity in the brain, our study provides information about the significant reaction of the cortical center to a single hand training session. Several studies have also reported similar results. For example, Kim et al. (2022) showed that RT with a single training session could induce a change in brain activity observed in fNIRs [9]. Vahdat et al. (2019) found that a single-training session of robot-assisted proprioceptive training in patients with chronic stroke could enhance reaching accuracy and alter functional sensorimotor connectivity [39]. To this end, although conventional rehabilitation applies a prolonged training period, the effect of a single training session can still be observed.

The present study has several limitations that could influence the interpretation of the results. First, precise marking of the timing of motor events is needed to explore motor task-related functional connectivity in different training sessions. Second, we only assessed the immediate effect of the intervention; future studies are needed to characterize the long-term therapeutic effects. Third, due to the sample size, we did not classify stroke patients according to functional motor severity, brain lesion location, brain lesion size, the severity of neurologic deficits, or whether rehabilitation training was promptly initiated. Fourth, we enrolled a limited number of patients with a wide range of stroke onset times, ranging from the acute to chronic stages. A larger sample size is needed to classify patients further according to their stroke onset time to validate our findings. Finally, functional imaging methods, such as fMRI, magnetoencephalography, and fNIRs, would shed light on activated cortical regions and connectivity in task-oriented training. Therefore, functional imaging should be applied in future studies to understand the neuronal mechanisms of functional recovery further.

## 5. Conclusions

We found that stroke duration influenced the effects of rehabilitation strategies on interhemispheric coherence. Further studies are needed to assess whether the results of interhemispheric coherence could be used as a biomarker to determine the optimal rehabilitation strategy for patients with varied post-stroke durations.

## Figures and Tables

**Figure 1 bioengineering-09-00727-f001:**
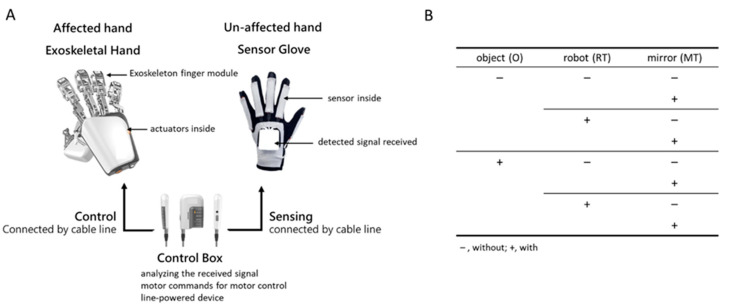
The exoskeleton hand device and the study design. (**A**) The exoskeleton hand device; (**B**) the diagram of 2 (with vs. without object manipulation) × 2 (with vs. without robot-assisted) × 2 (with vs. without mirror) factorially experimental design.

**Figure 2 bioengineering-09-00727-f002:**
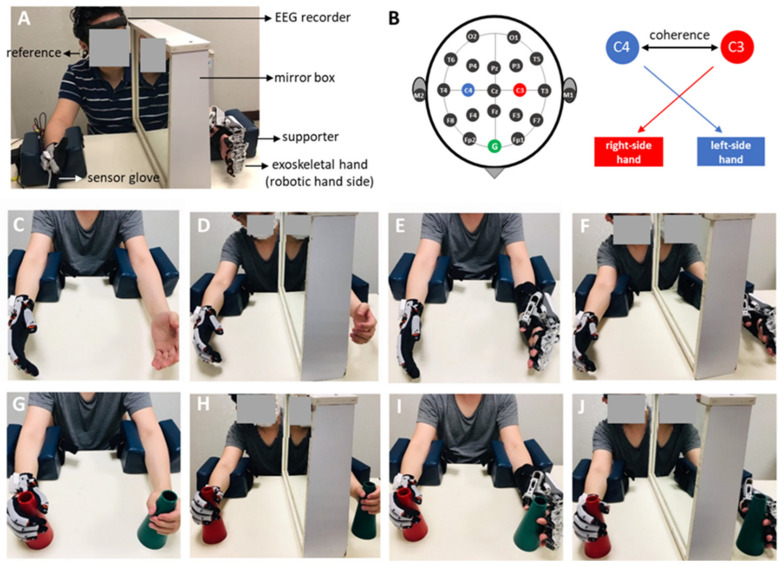
The experimental setup, analyzed electrodes, and the eight experimental conditions. (**A**) The experimental setup using the dry-electrode-based EEG recorder when the exoskeleton robot was applied on the participant. (**B**) The 10–20 EEG map, among which the coherence between C3 and C4 electrodes was analyzed, by assuming the right and left heads were controlled by the motor cortexes overlying the C3 and C4 electrodes, respectively. (**C**–**J**) The eight experimental conditions (using a factorial design of 2 object manipulation × 2 robot therapy × 2 mirror therapy conditions), including the unilateral hand movement without applying robot, mirror and object (**C**), conventional mirror therapy (**D**), robot-assisted bimanual training (**E**); robot-assisted mirror therapy (**F**); (**G**–**J**) were the trainings of (**C**–**F**) with object manipulation.

**Figure 3 bioengineering-09-00727-f003:**
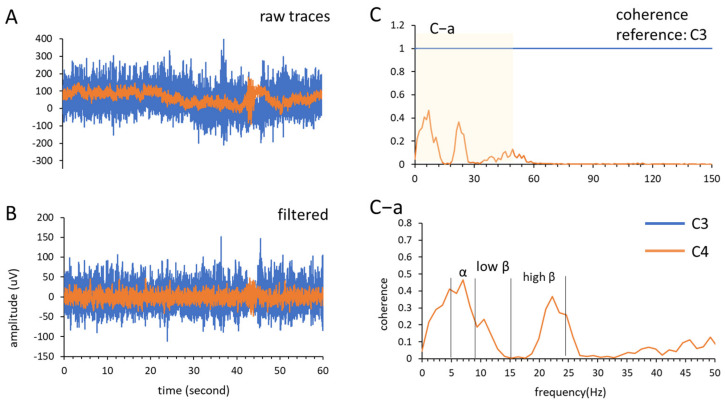
An example of MT × RT condition for coherence sum analysis. (**A**) The raw traces of channel C3 and C4 recoded from stroke patient under MT × RT condition; (**B**) the recorded traces were filtered (pass frequency band: 1–12-Hz); (**C**) coherence analysis using channel C3 as reference, the frequency band range in (**C**–**a**) was divided into three frequency bands and calculated the coherence sum.

**Figure 4 bioengineering-09-00727-f004:**
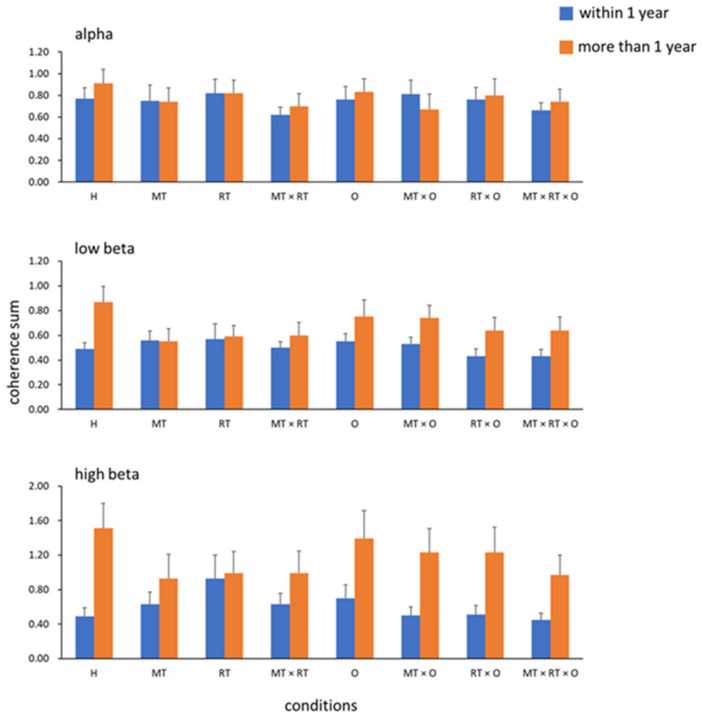
Interhemispheric coherences in the three bands: the alpha, low beta, and high beta bands.

**Table 1 bioengineering-09-00727-t001:** The demographic and clinical characteristics of the stroke patients.

Variable	Stroke Patient (n = 40)
sex (n)	
Male/Female	28/12
age (y)	53.28 ± 12.23
affected hand side (n)	
Right-side/Left-side	19/21
affected brain area ^1^ (n)	
cortical/sub-cortical	17/23
time elapse after stroke (m)	20.51 ± 23.41

^1^ Cortical area means the affected areas involving cortex; sub-cortical area including brainstem.

**Table 2 bioengineering-09-00727-t002:** Interhemispheric coherence sum in the onset ≤1 year group.

n = 20	Alpha		Low Beta		High Beta	
Effects	Mean	SEM	Mean	SEM	Mean	SEM
Hand	0.77	0.10	0.49	0.05	0.49	0.10
Mirror	0.75	0.15	0.56	0.08	0.63	0.14
Robot	0.82	0.13	0.57	0.13	0.93	0.27
Mirror × Robot	0.62	0.07	0.50	0.05	0.63	0.13
Object	0.76	0.12	0.55	0.06	0.70	0.15
Mirror × Object	0.81	0.13	0.53	0.06	0.50	0.10
Robot × Object	0.76	0.11	0.43	0.06	0.51	0.11
Mirror × Robot × Object	0.66	0.07	0.43	0.06	0.45	0.08

**Table 3 bioengineering-09-00727-t003:** Main effect analysis in the onset ≤1 year group. * *p* < 0.05.

n = 20	Alpha			Low Beta			High Beta		
Effects	F	*p*-Value	ⴄ^2^	F	*p*-Value	ⴄ^2^	F	*p*-Value	ⴄ^2^
Object	0.08	0.780	0.00	1.67	0.212	0.08	2.91	0.104	0.13
Robot	1.43	0.247	0.07	1.15	0.297	0.06	0.30	0.590	0.02
Mirror	2.99	0.100	0.14	0.03	0.874	0.00	1.17	0.293	0.06
Robot × Object	0.11	0.741	0.01	2.26	0.149	0.11	8.11	**0.010 ***	0.30
Mirror × Object	0.92	0.350	0.05	0.03	0.868	0.00	0.20	0.661	0.01
Robot × Mirror	1.42	0.249	0.07	0.30	0.588	0.02	0.52	0.481	0.03
Robot × Mirror × Object	0.04	0.841	0.00	0.69	0.417	0.03	2.27	0.148	0.11

**Table 4 bioengineering-09-00727-t004:** Interhemispheric coherence sum in the onset >1 year group.

n = 20	Alpha		Low Beta		High Beta	
Effects	Mean	SEM	Mean	SEM	Mean	SEM
Hand	0.91	0.13	0.87	0.13	1.51	0.29
Mirror	0.74	0.13	0.55	0.10	0.93	0.28
Robot	0.82	0.12	0.59	0.09	0.99	0.25
Mirror × Robot	0.70	0.11	0.60	0.11	0.99	0.26
Object	0.83	0.12	0.75	0.14	1.39	0.33
Mirror × Object	0.67	0.14	0.74	0.10	1.23	0.28
Robot × Object	0.80	0.15	0.64	0.10	1.23	0.29
Mirror × Robot × Object	0.74	0.12	0.64	0.11	0.97	0.23

**Table 5 bioengineering-09-00727-t005:** Main effect analysis in the onset >1 year group. * *p* < 0.05.

n = 20	Alpha			Low Beta			High Beta		
Effects	F	*p*-Value	ⴄ^2^	F	*p*-Value	ⴄ^2^	F	*p*-Value	ⴄ^2^
Object	0.36	0.553	0.02	1.33	0.26	0.07	2.18	0.156	0.10
Robot	0.21	0.654	0.01	2.86	0.11	0.13	4.88	0.040 *	0.20
Mirror	7.61	0.012 *	0.29	2.91	0.10	0.13	5.13	0.035 *	0.21
Robot × Object	1.54	0.230	0.07	0.02	0.89	0.00	0.02	0.888	0.00
Mirror × Object	0.31	0.587	0.02	3.19	0.09	0.14	0.54	0.470	0.03
Robot × Mirror	0.95	0.341	0.05	3.02	0.10	0.14	1.06	0.317	0.05
Robot × Mirror × Object	0.11	0.746	0.01	3.86	0.06	0.17	4.48	0.048 *	0.19

## Data Availability

The datasets generated and analyzed during the current study are available from the corresponding authors upon reasonable request. All data needed to evaluate the conclusions are presented in the paper.

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
