# Peer review of "Bilateral Sensorimotor Cortical Communication Modulated by Multiple Hand Training in Stroke Participants: A Single Training Session Pilot Study"

_bioengineering, 2022, doi:10.3390/bioengineering9120727_

Round 1

Reviewer 1 Report

Motor function impairment is a common outcome of stroke. The ‘over training’ programme and the recognized observation that it can induce a plastic expansion of the motor representation through practiced movements in the primary motor cortex is often applied to the rehabilitation of individuals recovering from brain injury (e.g., stroke). For example, Constraint-induced movement therapy (CIMT) involves the intensive use of the impaired limb while restraining the unaffected limb.  This is widely used to improve limb function after stroke. 

Therefore, the topic of this paper is relevant, timely, and of interest to the audience of this journal as it deals with rehabilitation training and functional recovery through cortical plasticity in stroke. In this preliminary pilot study, select stroke patients with different times of stroke duration, showed a significantly different degree of inter-hemispheric coherence that is thought to influence hand rehabilitation when subject to different tasks, based on EEG recordings. More specifically, Mirror Therapy (MT) and Robot Assisted Bimanual Therapy (RT) that were applied in this study showed significant differences in the bilateral training that emerged during training. Based on the suggestion that a desynchronized EEG may reflect an independent or desychronized neuronal state of networking, it is proposed that the index of interhemispheric communication can be a good suggestive indicator to take advantage of the best rehabilitation strategies to be adopted during stroke recovery.

In my opinion, the manuscript is technically correct and the methods are used correctly and clearly described, so that it is in my opinion that the data is sufficient to corroborate the claims. The reporting appears to also be sufficiently transparent to repeat the experiments for those not familiar with these types of studies. The graphs and figures with captions are appropriate and easy to comprehend and the supporting evidence and statistical data is reliable and properly validated.I also agree with the conclusions that were drawn up in this study based on the results presented. The manuscript should also be revised for proper grammar and written english

However, one main drawback of these experiments that might impact the results coming out from this study is the low number of recruited patients. This would undoubtedly impact the general trend and overall results as a result of the low statistical power.

Functional activation studies such as fMRI (for example) would have been very useful when applied, to give an insight on the functional level of the activated cortical region in performing a specific task over a broader field of view (rather than EEG). This could then be used to further the classification within and from the two groups based on differences based on activity-dependent plasticity changes as observed in the rehabilitation programmes that were performed. Functional neuroimaging and brain mapping suggest that the recovery of motor function after stroke leads to brain-wide modifications in neuronal activity patterns and connectivity. While initially tissue function and neurophysiological responses are diminished within the injured primary neocortex, cortical function increases over time. The introductory part of this manuscript should seek to talk about the documented literature that deals with the molecular mechanisms that are unmasked as a result of the increased cortical excitability in cortical regions distant from, but connected to the stroke core and the somatotopic reorganization that ensues.

One other drawback of this study is that the stroke patients were not classified according to the severity of the ictus. Some would have recovered to a better extent than others depending on the location of the infarct, the severity of the stroke, its size and whether physiotherapy (or other therapeutic strategy) was initiated in a timely fashion.  In human patients, the largest improvement occurs in the first 30 days after stroke, though significant progress is still found in patients with more severe deficits up to 90 days after stroke 

All these influential factors should be acknowledged in this study as they could influence the interpretation of the results. For a pilot experiment and in view that the authors are willing to discuss these further limitations in their studies (including any future work), I would accept this paper for publication.

Line 277 – ‘ever’ should read as ‘never’

Author Response

    We are very grateful for your constructive suggestions for improving the manuscript. We agree that the grammar needs to be improved, the limitation needs to be discussed more, and the reference needs to be updated. Accordingly, we have revised the manuscript point-by-point regarding the issues raised by the reviewers. We believe that the revised version would be more suitable for Bioenginnering.

Review #1

Motor function impairment is a common outcome of stroke. The ‘over training’ programme and the recognized observation that it can induce a plastic expansion of the motor representation through practiced movements in the primary motor cortex is often applied to the rehabilitation of individuals recovering from brain injury (e.g., stroke). For example, Constraint-induced movement therapy (CIMT) involves the intensive use of the impaired limb while restraining the unaffected limb.  This is widely used to improve limb function after stroke. 

Therefore, the topic of this paper is relevant, timely, and of interest to the audience of this journal as it deals with rehabilitation training and functional recovery through cortical plasticity in stroke. In this preliminary pilot study, select stroke patients with different times of stroke duration, showed a significantly different degree of inter-hemispheric coherence that is thought to influence hand rehabilitation when subject to different tasks, based on EEG recordings. More specifically, Mirror Therapy (MT) and Robot Assisted Bimanual Therapy (RT) that were applied in this study showed significant differences in the bilateral training that emerged during training.

Based on the suggestion that a desynchronized EEG may reflect an independent or desychronized neuronal state of networking, it is proposed that the index of interhemispheric communication can be a good suggestive indicator to take advantage of the best rehabilitation strategies to be adopted during stroke recovery.

In my opinion, the manuscript is technically correct and the methods are used correctly and clearly described, so that it is in my opinion that the data is sufficient to corroborate the claims. The reporting appears to also be sufficiently transparent to repeat the experiments for those not familiar with these types of studies. The graphs and figures with captions are appropriate and easy to comprehend and the supporting evidence and statistical data is reliable and properly validated. I also agree with the conclusions that were drawn up in this study based on the results presented.

Q1: The manuscript should also be revised for proper grammar and written English.

Response:

Thanks for the suggestion. We have revised the grammar and written English in the present manuscript. The appendix shows the proof of English editing.

Q2: However, one main drawback of these experiments that might impact the results coming out from this study is the low number of recruited patients. This would undoubtedly impact the general trend and overall results as a result of the low statistical power.

Response:

Thanks for the comment. We agree that the sample size is relatively low in the present work. We noted this limitation and discussed in the Discussion section, as below (line 342-347):

“ Third,…..A larger sample size is needed in the future study to validate our findings.”

Q3: Functional activation studies such as fMRI (for example) would have been very useful when applied, to give an insight on the functional level of the activated cortical region in performing a specific task over a broader field of view (rather than EEG). This could then be used to further the classification within and from the two groups based on differences based on activity-dependent plasticity changes as observed in the rehabilitation programmes that were performed. Functional neuroimaging and brain mapping suggest that the recovery of motor function after stroke leads to brain-wide modifications in neuronal activity patterns and connectivity.

Response:

Thanks for the comment. We agree that functional activation studies, such as fMRI, would have been very useful to give an insight on the functional level of the activated cortical region in performing a specific task over a broader field of view than EEG. However, the functional image cannot be applied at rehabilitation wards or clinics and the availability for clinical application is less than EEG. Therefore, we did not choose fMRI as the assessment tool in our study. We added the potential of functional neuroimaging at the end of Discussion section, as below (line 347-351):

“Finally, Functional imaging methods, …. of functional recovery further.

Q4: While initially tissue function and neurophysiological responses are diminished within the injured primary neocortex, cortical function increases over time. The introductory part of this manuscript should seek to talk about the documented literature that deals with the molecular mechanisms that are unmasked as a result of the increased cortical excitability in cortical regions distant from, but connected to the stroke core and the somatotopic reorganization that ensues.

Response:

Thanks for your constructive comments. We have revised the 4th paragraph in the Introduction by including studies regarding interhemispheric corticocortical interaction, as below (line 54-66):

“Regarding functional imaging that investigated interhemispheric activation, Christian Grefke and Gereon R Fink (2014) …. as participants receive training.”

Q5: One other drawback of this study is that the stroke patients were not classified according to the severity of the ictus. Some would have recovered to a better extent than others depending on the location of the infarct, the severity of the stroke, its size and whether physiotherapy (or other therapeutic strategy) was initiated in a timely fashion.  In human patients, the largest improvement occurs in the first 30 days after stroke, though significant progress is still found in patients with more severe deficits up to 90 days after stroke 

Response:

Thanks for the comment. We agree it would be useful to further understand the brain functional recovery by classifying the stroke patient according to the severity of the ictus, the brain lesion location, brain lesion size, the severity of neurologic deficits, and whether rehabilitation training was initiated in a timely fashion. Therefore, we added these factors in the limitation part of Discussion, as below (line 342-346):

“Third, due to the sample size….. from the acute to chronic stages. ”

All these influential factors should be acknowledged in this study as they could influence the interpretation of the results. For a pilot experiment and in view that the authors are willing to discuss these further limitations in their studies (including any future work), I would accept this paper for publication.

 Q6: Line 277 – ‘ever’ should read as ‘never’

Response:

Thanks for the comment. To make the sentence clearer we have corrected the word to “never”.

Reviewer 2 Report

Authors attempt to find out „the neural substrate” (of the neuroplasticity?) during the hand training with Bi-manual therapy (BT), mirror therapy (MT), and robot-assisted rehabilitation in patients after stroke.

The aim is brave and the attempt is significant for the reader related to the problems of the cortical neuroplasticity phenomenon and its current concepts still not explained to the end.

Bilateral sensorimotor cortical communication with EEG was studied when participants were performing hand tasks under different intervention conditions. A sum of spectral coherence was applied to analyse the C3 and C4 signals to measure the level of bilateral cortical communication.

Stroke patients from the acute to the chronic stage were enrolled in this study, and neuroplasticity processes are known to appear in such cases differently, duration and their effects are recorded and fixed differently. This in fact the first result and conclusion of the Authors from their study. Moreover, EEG results provided evidence that the stroke duration might influence the hand rehabilitation effects on interhemispheric coherence.

Summarizing above, the results confirmed the well-known course of hand rehabilitation in post-stroke patients using EEG as the research tool, but the algorithm of EEG analysis is creative and brings new relevance to the study of the human brain neuroplasticity processes.

The question is if and how long the positive effects of rehabilitation will survive, but the Authors promise long-lasting observations.

Minor

Maybe the term „the mechanisms of actions” instead of „the neural substrate” would sound better in the Abstract and in the Introduction section (lines 43-44). The authors do not perform the structural neuroimaging studies but the functional ones, which in the case of cortical interactions are explained with the neurophysiological method.

The Introduction is well written, and state-of-art nicely presented, however, some of the cited articles although precisely selected are old and need upgrading with the two-three newest, of course except the historical Jasper and Penfield poem. It does not refer to the EEG findings in this topic but the development and achievements of the neurorehabilitation methods of post-stroke patients.

M&M is OK except for some phrases which need more precise explaining eg. (line 71) “…(2) first stroke patient diagnosed by clinical and imaging findings; (line 102)…”, “… The electrodes of interest were C3 and C4,…” (you mean …Particularly monitored C3 and C4 recordings” or “C3 and C4 marker electrodes”?)- unclear, in the realm of guesswork, not every reader is the neurologist or the neurophysiologist. The principles of the applied methods in the therapeutic session are very clearly presented.

The number of patients should be mentioned in the Abstract and in 2.1. Participants and Design of M&M section. How many were acute or chronic?

Brunnstrom recovery stages for evaluation of stroke patients should be explained with the ref. in M&M section.

The result of this scale is mentioned in the Results section. It is not mentioned in the Abstract, but important for clinicians.

Apart from the above the Results section is well written.

The content of Discussion in the most of discussed topics requires applying the old literature but upgrading with the newest would be cordially invited (minor suggestion). In my opinion, the significant reaction of the cortical centers to the single training session should be underlined and broader explained.

Stroke patients from the acute to the chronic stage were enrolled in this study, is this not a limitation?

The names of journals in references are sometimes different with the big-small fashion of letters (eg. Journal of neurophysiology vs Archiv für Psychiatrie und Nervenkrankheiten, the most striking example is Acta neurobiologiae experimentalis, line 395). Ref. 33 has no year and volume.

Author Response

        We are very grateful for your constructive suggestions for improving the manuscript. We agree that the grammar needs to be improved, the limitation needs to be discussed more, and the reference needs to be updated. Accordingly, we have revised the manuscript point-by-point regarding the issues raised by the reviewers. We believe that the revised version would be more suitable for Bioenginnering.

Review #2

Q1: Authors attempt to find out „the neural substrate” (of the neuroplasticity?) during the hand training with Bi-manual therapy (BT), mirror therapy (MT), and robot-assisted rehabilitation in patients after stroke.

Response:

Thanks for the comment. We have replaced the term “the neural substrate” with “the mechanisms of actions” according to your suggestion in Q3.

The aim is brave and the attempt is significant for the reader related to the problems of the cortical neuroplasticity phenomenon and its current concepts still not explained to the end.

Bilateral sensorimotor cortical communication with EEG was studied when participants were performing hand tasks under different intervention conditions. A sum of spectral coherence was applied to analyse the C3 and C4 signals to measure the level of bilateral cortical communication.

Stroke patients from the acute to the chronic stage were enrolled in this study, and neuroplasticity processes are known to appear in such cases differently, duration and their effects are recorded and fixed differently. This in fact the first result and conclusion of the Authors from their study. Moreover, EEG results provided evidence that the stroke duration might influence the hand rehabilitation effects on interhemispheric coherence.

Summarizing above, the results confirmed the well-known course of hand rehabilitation in post-stroke patients using EEG as the research tool, but the algorithm of EEG analysis is creative and brings new relevance to the study of the human brain neuroplasticity processes.

Q2: The question is if and how long the positive effects of rehabilitation will survive, but the Authors promise long-lasting observations.

 Response:

As it is a one-session pilot study, we did not know how long the positive effects of rehabilitation will survive. Therefore, a long-lasting observation study should be perform in the future to characterize this issue. Therefore, we mentioned this limitation in the Discussion, as below (line 340-342 ):

Second, we only assessed the immediate effect of the intervention; future studies are needed to characterize the long-term therapeutic effects.

Minor

Q3: Maybe the term „the mechanisms of actions” instead of „the neural substrate” would sound better in the Abstract and in the Introduction section (lines 43-44). The authors do not perform the structural neuroimaging studies but the functional ones, which in the case of cortical interactions are explained with the neurophysiological method.

 Response:

Thanks for the suggestion, indeed, the usage of the term “the mechanisms of actions” would be better than “neural substrate” in the context. Thus, we have changed the terms in the Abstract and Introduction sections.

Q4: The Introduction is well written, and state-of-art nicely presented, however, some of the cited articles although precisely selected are old and need upgrading with the two-three newest, of course except the historical Jasper and Penfield poem. It does not refer to the EEG findings in this topic but the development and achievements of the neurorehabilitation methods of post-stroke patients.

 Response:

Thanks for the comment. We have revised the paragraph as the response for the Q4 in the reviewer 1’ comment, as below (line 54-66):

“Regarding functional imaging that investigated interhemispheric activation, Christian Grefke and Gereon R Fink (2014) …. as participants receive training.”

Q5: M&M is OK except for some phrases which need more precise explaining eg. (line 71) “…(2) first stroke patient diagnosed by clinical and imaging findings; (line 102)…”, “… The electrodes of interest were C3 and C4,…” (you mean …Particularly monitored C3 and C4 recordings” or “C3 and C4 marker electrodes”?)- unclear, in the realm of guesswork, not every reader is the neurologist or the neurophysiologist. The principles of the applied methods in the therapeutic session are very clearly presented.

Response:

Thanks for the suggestions. We have revised the uncleared phrases as below:

  1. Inclusion criteria (2) (line 79-80):

“first-ever stroke patients diagnosed by clinical presentation and imaging findings;”

  1. EEG recording (line 119-121):

“To investigate bilateral sensorimotor cortical activities, the recording channels overlying the primary sensorimotor and mesial motor areas, marked by C3 and C4, respectively, were used for analysis”

Q6: The number of patients should be mentioned in the Abstract and in 2.1. Participants and Design of M&M section. How many were acute or chronic?

Response:

Thanks for the suggestion. In our experimental design, we did not set the participant number in every stroke onset period. We enrolled the participants according to our inclusion and exclusion criteria. Thus, we revised the results in the Abstract as below (line 22-24):

“(1) We included stroke patients with…. Brunnstrom recovery stage ranged from 2 to 4 

Q7: Brunnstrom recovery stages for evaluation of stroke patients should be explained with the ref. in M&M section.

Response:

Thanks for the suggestion, we have added the information about Brunnstrom recovery stages assessment in the Participants and Design section in the M&M as below (line 85-88):

“We applied Brunnstorm recovery stages …..on stroke patient assessment [19].”

Q8: The result of this scale is mentioned in the Results section. It is not mentioned in the Abstract, but important for clinicians. Apart from the above the Results section is well written.

 Response:

Thanks for the suggestion, we have included the information about Brunnstrom recovery stage in the Abstract. Please find the response in the Q5 above.

Q9: The content of Discussion in the most of discussed topics requires applying the old literature but upgrading with the newest would be cordially invited (minor suggestion). In my opinion, the significant reaction of the cortical centers to the single training session should be underlined and broader explained.

Response:

Thanks for the comments. We agree with the effect of a single-training session should be discussed more. Thus, we revised the sentence and added a paragraph in the Discussion section, as below (line 321-325):

“A functional near-infrared spectroscopy (fNIRs)….. supporting our findings.”

& (line 329-337)

“Although a single training session was not considered ….the effect of single training can still be observed.”

Q10: Stroke patients from the acute to the chronic stage were enrolled in this study, is this not a limitation?

Response:

Thanks for the suggestion. We agree that the other limitation in our study is that we enrolled patients with a wide range of stroke onset time from the acute to the chronic stage. Therefore, we revise the limitation in Discussion, as below (line 345-347):

“Fourth, we enrolled a limited number of patients with a wide range of stroke onset times, ranging from the acute to chronic stages. A larger sample size is needed to classify patients further according to their stroke onset time to validate our findings” 

Q11: The names of journals in references are sometimes different with the big-small fashion of letters (eg. Journal of neurophysiology vs Archiv für Psychiatrie und Nervenkrankheiten, the most striking example is Acta neurobiologiae experimentalis, line 395). Ref. 33 has no year and volume.

Response:

Thanks for the comment. We have corrected the style of reference in the present manuscript.
